# *FAK*-Copy-Gain Is a Predictive Marker for Sensitivity to FAK Inhibition in Breast Cancer

**DOI:** 10.3390/cancers11091288

**Published:** 2019-09-02

**Authors:** Young-Ho Kim, Hyun-Kyoung Kim, Hee Yeon Kim, HyeRan Gawk, Seung-Hyun Bae, Hye Won Sim, Eun-Kyung Kang, Ju-Young Seoh, Hyonchol Jang, Kyeong-Man Hong

**Affiliations:** 1Research Institute, National Cancer Center, Goyang 10408, Korea; 2Department of Pharmacology and New Drug Development Research Institute, Chonbuk National University Medical School, Jeonju 54689, Korea; 3Department of Cancer Biomedical Science, National Cancer Center Graduate School of Cancer Science and Policy, Goyang 10408, Korea; 4Departments of Microbiology, Ewha Womans University School of Medicine, Ewha Medical Research Center, Seoul 07804, Korea

**Keywords:** copy gain of *FAK*, FAK inhibitor, FAK-knockdown, target therapy, breast cancer, AKT signaling

## Abstract

Background: Cancers with copy-gain drug-target genes are excellent candidates for targeted therapy. In order to search for new predictive marker genes, we investigated the correlation between sensitivity to targeted drugs and the copy gain of candidate target genes in NCI-60 cells. Methods: For eight candidate genes showing copy gains in NCI-60 cells identified in our previous study, sensitivity to corresponding target drugs was tested on cells showing copy gains of the candidate genes. Results: Breast cancer cells with *Focal Adhesion Kinase* (*FAK*)-copy-gain showed a significantly higher sensitivity to the target inhibitor, FAK inhibitor 14 (F14). In addition, treatment of F14 or FAK-knockdown showed a specific apoptotic effect only in breast cancer cells showing *FAK*-copy-gain. Expression-profiling analyses on inducible *FAK* shRNA-transfected cells showed that FAK/AKT signaling might be important to the apoptotic effect by target inhibition. An animal experiment employing a mouse xenograft model also showed a significant growth-inhibitory effect of F14 on breast cancer cells showing *FAK*-copy-gain, but not on those without *FAK*-copy-gain. Conclusion: *FAK*-copy-gain may be a predictive marker for FAK inhibition therapy in breast cancer.

## 1. Introduction

Cancer remains an incurable disease worldwide with a high mortality rate, owing, to a significant extent, to its unresponsiveness or resistance to chemotherapeutic agents. There are several predictive markers that are employed to select candidate cancer patients for targeted therapy. For enhanced cancer patient survival, still more markers are needed. Copy gains of drug-target genes have proved the most important predictive markers for chemotherapy in cancer patients. For example, *ERBB2*-copy-gain is a predictive marker for anti-ERBB2 inhibitor treatment in breast [1] and gastric adenocarcinomas [2]. *EGFR*-copy-gain for anti-EGFR inhibitors [3,4], *MET*-copy-gain for MET tyrosine kinase inhibitors [5,6], and *FGFR2*-copy-gain for FGFR2 phosphorylation inhibitors [7] are other examples. Copy gain can also be a marker for chemoresistance, as in *MET*-copy-gain for Gefitinib resistance in lung cancer [8] or copy gains of *LAPTM4B* and *YWHAZ* for anthracycline resistance in breast cancer [9]. Therefore, identification of novel copy-gain predictive markers in cancer is essential for effective selection of candidate patients who might respond better to specific targeted agents.

We have reported copy-gain genes in NCI-60 cells employing a modified real competitive PCR (mrcPCR) method [10]. In that study, we found that eight drug-target genes including *FAK* (or *PTK2*), *MYC*, *EGFR*, *ERBB2*, *FGFR1*, *MET*, *IGF1R*, and *MAP2K2* gained copies in NCI-60 cells. It is a generally accepted notion that amplified drug-target genes are excellent predictive markers for chemotherapy in cancer. Therefore, we investigated, by treatment of targeted drugs to NCI-60 cancer cells showing copy-gain drug-target genes, whether the copy gains of the eight candidate drug-target genes can be predictive markers for chemotherapy.

In our analyses, we found that breast cancer cells with *FAK*-copy-gain showed higher sensitivity to FAK inhibitor 14 (F14), a FAK inhibitor, which is known to inhibit cancer cell growth via its inhibition of FAK phosphorylation at Y397 [11]. FAK, Focal Adhesion Kinase, is a nonreceptor tyrosine kinase which plays significant roles in survival and invasion of cancer cells, and its higher expression in many types of cancers, which is implicated in cancer survival, has been reported [12,13,14]. In the present study, FAK knockdown and mouse xenograft studies were performed on breast cancer cells with and without *FAK*-copy-gain to investigate the possibility of *FAK*-copy-gain as a predictive marker in breast cancer.

## 2. Results

### 2.1. Drug Sensitivities in Nci-60 Cells with Copy-Gain Drug-Target Genes

It is generally thought that cancer cells with copy-gain drug-target genes are sensitive to targeted drugs. To investigate the correlation, the relative drug sensitivities were determined in NCI-60 cells showing copy-gain drug-target genes (Figure 1A). In our previous study on copy number gain of drug-target genes in NCI-60 cells [10], eight genes, namely *FAK*, *MYC*, *EGFR*, *ERBB2*, *FGFR1*, *MET*, *IGF1R*, and *MAP2K2*, showed copy gain in NCI-60 cells. In the present study, we treated specific targeted drugs to cells showing copy gain of the corresponding target genes. We found that *FAK*-copy-gain showed a significant correlation with higher sensitivity to its targeted drug, F14, in breast and ovarian cancer cell lines (*p* = 0.001, Figure 1B,C). However, the correlation between *FAK*-copy-gain and the sensitivity to F14 was not observed in cells of other tissue origin (e.g., lung cancer), as shown in Figure 1D (*p* = 0.999).

In the other seven candidate genes apart from *FAK*, the correlation between copy gain and target-drug sensitivity was not significant (Appendix A). Copy-gain cells for *ERBB2*, *MET*, *IGF1R*, and *MAP2K2* numbered only 1 or 2 among the NCI-60 cells. Therefore, although the estimation of correlation might not be adequate, the sensitivity to targeted drugs in cells showing copy gains for those four genes was quite similar to those from other cells with no copy gain. In one breast cancer cell with high amplification of *EGFR*, sensitivity to its target drug, Gefitinib, was much higher than that in the other breast cancer cells, suggesting a possible correlation between *EGFR*-copy-gain and sensitivity to Gefitinib. Among cells with *MYC*-copy-gain, however, there was no significant difference in the sensitivity to its target inhibitor, MYC inhibitor II (*p* = 0.384, Figure 1E), although *MYC*-copy-gain was observed in as many as 13 of the NCI-60 cells.

### 2.2. Significantly Higher Sensitivity to Fak Inhibition in Breast Cancer Cells with FAK-Copy-Gain

There were not enough breast cancer cells among the NCI-60 cells to evaluate the significance of the correlation of *FAK*-copy-gain with sensitivity to F14, because the correlation was significant only when the data for ovarian cancer cells were combined. The correlation in breast cancer alone was not clear, and so we treated F14 to an additional 10 breast cancer cells. After mrcPCR determination of *FAK* copies for the additional breast cancer cells, we again tested the statistical significance of the correlation between *FAK*-copy-gain and F14 sensitivity in breast cancer cells. In a total of 16 breast cancer cells, including seven cells showing *FAK*-copy-gain (Figure 2A), the correlation was significant (*p* < 0.001, Figure 2B), suggesting that *FAK*-copy-gain might be a significant predictive marker for sensitivity to F14 in breast cancer cells.

To evaluate the level of *FAK* RNA and its protein in breast cancer cells with *FAK*-copy-gain, RT-PCR and Western blotting analyses were performed. The level of *FAK* RNA expression was significantly higher in cells with *FAK*-copy-gain than in those without (*p* < 0.001, Figure 2C), which is consistent with the results from the Cancer Cell Line Encyclopedia (CCLE) database, as shown in Appendix A. Consistent with RNA expression, the FAK protein level was also higher in copy-gain cells (Figure 2D), suggesting that *FAK*-copy-gain might have biological effects on enhanced expression of FAK in breast cancer cells. The higher expression of FAK in *FAK*-copy-gain cancer tissues has also been reported previously [15].

A significant correlation between *FAK*-copy-gain and F14 sensitivity is a potentially important predictive marker for breast cancer patients, and the number of candidate cancer patients might be important as well. From our review of cases from the TCGA and METABRIC datasets [16], about 15% of breast cancer cases show *FAK*-copy-gain (Figure 2D), suggesting that treatment with FAK inhibitors might be beneficial for a significant fraction of breast cancer patients.

### 2.3. Apoptotic Effect of F14 in Breast Cancer Cells With FAK-Copy-Gain

To investigate the effect of FAK inhibitors in breast cancer cells with *FAK*-copy-gain, the relative level changes of activated FAK or phosphorylated FAK (pFAK) at Y397 were monitored after F14 treatment. On Western blot analysis, pFAK, in contrast to the relatively stable expression of FAK, decreased after F14 treatment in *FAK*-copy-gain breast cancer cells such as BT-549 and MDA-MB-453 (Figure 3A), indicating that F14 inhibits the activation of FAK.

The level of cleaved caspase-3, a marker of apoptosis, increased after F14 treatment in *FAK*-copy-gain cells (BT-549 and MDA-MB-453), in contrast to the nondetection of cleaved caspase-3 in cells without *FAK*-copy-gain (MDA-MB-231 and JIMT-1), as shown in Figure 3B, suggesting a specific apoptotic effect of F14 on *FAK*-copy-gain cells. To confirm this result, a flow-cytometric analysis employing fluorophore-conjugated annexin V was performed (Figure 3D): after F14 treatment, the *FAK*-copy-gain cells showed significantly more annexin V-positive cells than the controls did (* in Figure 3C, *p* < 0.001), but those without *FAK*-copy-gain did not (not significant, ns). These data suggest that the apoptotic effect of FAK inhibitor is specific to breast cancer cells with *FAK*-copy-gain.

To further confirm the specific apoptotic effect of FAK inhibition on *FAK*-copy-gain cells, expression of *FAK* shRNA was induced after transfection of doxycycline-inducible *FAK* shRNA-vector to breast cancer BT-549 (*FAK*-copy-gain) and MDA-MB-231 (no *FAK*-copy-gain) cells. Induction of *FAK* shRNA by doxycycline reduced the FAK level, which results in pFAK reduction in both cell lines. However, caspase-3 cleavage was observed only in the BT-549 cells (Figure 3E,F), which is consistent with the results by F14 treatment. In the colony-formation assay, a significant reduction in colony formation was observed in the BT-549 cells (*p* = 0.002, Figure 3G), in contrast to the nonsignificant reduction in the MDA-MB-231 cells (*p* = 0.121). These results suggest that the apoptotic effect by FAK inhibition is more effective in *FAK*-copy-gain breast cancer cells.

### 2.4. Downstream Molecular Changes by FAK-Knockdown in FAK-Copy-Gain Cells

To further investigate the reasons for the higher apoptotic effect on *FAK*-copy-gain cells, we analyzed expression profiles before and after induction of *FAK* shRNA in breast cancer cells such as BT-549 (copy gain) and MDA-MB-231 (no copy gain) by total RNA sequencing. In the BT-549 cells, there were a lot of differentially expressed genes (DEGs) after *FAK* shRNA induction (Figure 4A, list of genes in Appendix A), and most of the DEGs were related to cell viability, migration, and movement (Figure 4B). In contrast, the DEG pattern of BT-549 by FAK-knockdown was quite different from that of MDA-MB-231 (Figure 4C, list of genes in Appendix A), suggesting that the difference might be related to the difference between cells with and without *FAK*-copy-gain in apoptotic effect by F14 treatment or FAK-knockdown. Therefore, the DEGs shown in BT-549 but not in MDA-MB-231 were analyzed again, and we found that most of the BT-549-specific DEGs were genes related to cell viability and contact growth inhibition, but not to cell migration or movement (Figure 4D). The individual BT-549-specific DEGs related to cell viability are shown in Figure 4E.

Among the BT-549-specific DEGs, we noted that many AKT-signaling molecules were changed by FAK-knockdown (Figure 4F), suggesting that AKT might be an important factor behind the preferential apoptotic process in *FAK*-copy-gain breast cancer cells. To investigate the possibility of the involvement of AKT signaling during the apoptotic process by FAK inhibition, the level of activated AKT or phosphorylated AKT (pAKT) was analyzed by Western blotting: pAKT at S473 decreased after induction of *FAK* shRNA in BT-549 cells (Figure 4G), whereas the level of pAKT was minimal at the basal level in MDA-MB-231 cells (Figure 4G), which might explain the specific effect of FAK downregulation only in breast cancer cells with *FAK*-copy-gain via AKT signaling.

To confirm the importance of AKT signaling in *FAK*-copy-gain breast cancer cells during the apoptotic process, Western blot analyses on cancer cells with and without *FAK*-copy-gain were performed before and after F14 treatment. In the *FAK*-copy-gain cells (BT-549 and MDA-MB-453), F14 treatment reduced the pAKT at S473, whereas the reduction of pAKT was minimal or reversed in cells with no *FAK*-copy-gain (MDA-MB-231 and JIMT1) (Figure 4H). Therefore, the specific growth-inhibition effect by F14 or *FAK* shRNA on only *FAK*-copy-gain cells is related to the intact FAK to AKT signaling (Figure 4I).

### 2.5. Effect of F14 On Tumor Progression in a Mouse Xenograft Model

To confirm the specific inhibitory effect of F14 on only breast cancer cells with *FAK*-copy-gain, two cell lines with *FAK*-copy-gain (BT-549 and MDA-MB-453) and two without copy gain (MDA-MB-231 and JIMT-1) were employed for a mouse xenograft model. F14 was injected 5 days a week for 4 weeks after the implantation of cancer cells (Figure 5A). A growth-inhibitory effect was evident only in *FAK*-copy-gain cells (*p* < 0.001 for MDA-MB-453 and BT-549 by two-way RM ANOVA, Figure 5B), not in cells without *FAK*-copy-gain (MDA-MB-231 and JIMT-1 with *p*-values of 0.110 and 0.783, respectively). Photographs for tumor explants from three representative mice for each group are shown in Appendix A. Our animal results also suggest that FAK inhibitors have an inhibitory effect on breast cancer cells with *FAK*-copy-gain but only a minimal effect on cells with no *FAK*-copy-gain, which is quite consistent with our in vitro results.

## 3. Discussion

In a search for predictive markers among eight candidate genes showing copy gain in NCI-60 cells, *FAK*-copy-gain was significantly correlated with enhanced drug-sensitivity to a FAK inhibitor, F14, in breast and ovarian cancer cells. Treatment of F14 or FAK-knockdown showed a specific apoptotic effect on breast cancer cells with *FAK*-copy-gain, but not on cells without *FAK*-copy-gain. The specific apoptotic effect on only *FAK*-copy-gain cells by F14 or FAK-knockdown was associated with the AKT-signaling process. Specificity in breast cancer cells with *FAK*-copy-gain was also observed in a mouse xenograft model. Therefore, our study suggests that *FAK*-copy-gain could be a significant predictive marker for sensitivity to FAK inhibition in breast cancer.

In cancer cells showing copy gains of the other seven genes apart from *FAK*, the correlation with sensitivity to each targeted inhibitor was not significant in the present study. Especially, *MYC*-copy-gain seems not to be a predictive maker of an MYC inhibitor. However, the present study’s finding of noncorrelation between copy gain and sensitivity to targeted inhibitors for some markers might be attributable to the fact that only a few cells among the NCI-60 cells showed copy gains for specific drug-target genes. For example, a cell showing high copy gain of *EGFR* had dramatically higher sensitivity to its target drug, Gefitinib, although it was not significant statistically. It should be noted that copy gains of *EGFR* [3,4], *FGFR1* [17], *MET* [5,6], and *ERBB2* [1,2] have already been reported to be predictive markers for certain tumor types. From our limited analyses on NCI-60 cells, however, only *FAK*-copy-gain was significantly associated with higher sensitivity to its targeted inhibitor, F14, at least in breast cancer cells; this indicates that *FAK*-copy-gain might be a significant molecular marker for targeted therapy with FAK inhibitors in breast cancer.

A past report suggested that growth inhibition by FAK inhibitors might be due to off-target effects, in that growth inhibition by a FAK inhibitor, PF-573,228, was observed at a 10-times-higher concentration than that for migration inhibition [18]. However, growth inhibition by F14 has already been reported [11], and indeed, the present study’s finding of higher growth inhibition and apoptosis by F14 or FAK knockdown in *FAK*-copy-gain cells than in those without copy gain, suggests that growth inhibition by F14 or FAK-knockdown is not an off-target effect but rather a FAK-specific effect. Consistent with our finding, a recent report on a new FAK inhibitor, BI853520, also showed that FAK inhibitor can have a clear growth-inhibitory effect in a 3D culture with a lesser drug concentration than that necessary for an antimigratory effect [19].

In the present study, the apoptotic effect of FAK inhibitors was observed only in *FAK*-copy-gain cells both in vitro and in mouse xenograft model systems, in which specificity appears to be related to AKT signaling, one of the key signals for apoptosis [20]. FAK has effects on both cell migration/invasion [21] and cell survival [22]. Especially for cell-survival effects in breast cancer cells, FAK inhibition by a dominant negative mutant drives the apoptotic process via Fas-associated death-domain protein and caspase-8 [23]. Many FAK inhibitors, including F14, inhibit phosphorylation at Y397, which is a binding site for SH2-domain-containing molecules including SRC and PI3K for both cell survival and migration/invasion [22,24,25]. The present study observed, from the analyses of FAK knockdown cells, downregulation of AKT-signaling molecules in a *FAK*-copy-gain cell, and additionally, inhibition of AKT phosphorylation by F14 was observed specifically in *FAK*-copy-gain cells. Because phosphorylated or activated AKT enhance survival and inhibit apoptosis by inactivation of target proteins, including the proapoptotic BAD and the tumor suppressor p53 [26,27], AKT is a potentially important FAK effector molecule for cancer cell survival. In line with this, FAK signaling to PI3K/AKT has already been reported [25,28], and indicates that AKT is a plausible target of FAK inhibition, especially for survival of *FAK*-copy-gain breast cancer cells.

Clinically, increased FAK expression is associated with worse prognosis in lung [29] and colon cancer [30], and *FAK*-copy-gain is one of the mechanisms of FAK overexpression in breast and gastric cancers [15,31]. The present study also showed that *FAK*-copy-gain cells led to more highly expressed FAK, suggesting again that *FAK*-copy-gain is possibly an important mechanism of FAK overexpression or activation. In a previous study, *FAK*-copy-gain was observed in 18% of breast cancer tissues [15], with higher FAK expression in the triple negative subgroup [14]; similarly, our analyses of TCGA and METABRIC breast cancer cases showed that over 15% of breast cancer cases had *FAK*-copy-gain, as is consistent with our finding of frequent *FAK*-copy-gains in NCI-60 cells, thereby suggesting that many breast cancer cases can be candidates for FAK-inhibitor chemotherapy.

Several FAK inhibitors have already been reported [11,18,19,32,33,34], and clinical trials on those and others have been completed or are still ongoing (clinicaltrials.gov). However, the most effective cancer type for FAK-inhibitor treatment is currently unknown. Studies even failed to show activity for a FAK inhibitor, GSK2256098, that was combined with trametinib in unselected pancreatic cancers [35,36]. Our present study showed a significant correlation between *FAK*-copy-gain and sensitivity to its targeted drug, F14, in breast cancer, suggesting that *FAK*-copy-gain can be an effective predictive molecular marker in the selection of candidate patients for treatment with FAK inhibitors.

## 4. Materials and Methods 

### 4.1. Cell Culture and Chemicals

NCI-60 cell lines were obtained from the National Cancer Institute (MTA No. 2702-09). Additional breast cancer cell lines were purchased from either the Korean Cell Line Bank (Seoul, Korea) for MDA-MB-453, HCC-1954, HCC-1937, BT-20, JIMT-1, HCC-1419, and HCC-38, or from the American Tissue Culture Collection (Manassas, VA, USA) for SKBR3, HCC-1569, and Au565.

The following targeted drugs were employed as targeted inhibitors: FAK inhibitor 14 (F14) for FAK, GW583340 for ERBB2, PHA665752 for MET, and PD0325901 for MAP2K2 from Tocris Bioscience (Bristol, UK); PD173074 for FGFR1 from Abcam (Cambridge, UK); c-MYC inhibitor II, c-MYC inhibitor for MYC, and Gefitinib for EGFR, and Podophyllotoxin for IGF1R from Sigma-Aldrich (St. Louis, MO, USA).

### 4.2. Determination of Half-Maximal Inhibitory Concentration (IC_50_)

Cell viability and proliferation were determined by the crystal violet method as already described [37]. Crystal violet (#V5265) was purchased from Sigma-Aldrich. The data were normalized to untreated controls. Experiments were performed in three independent assays, each with three replicates. The dose–response curve was plotted using a non-linear regression model, and the inhibitory concentration (IC_50_) was determined from the fitted curves using GraphPad Prism version 5 (GraphPad Software Inc., San Diego, CA, USA).

### 4.3. Colony-Formation Assay

A colony-formation assay was performed according to a method described previously in [38]. Briefly, 500 cells per well in a six-well plate were incubated with or without doxycycline (1 µg/mL). After 15 days of incubation, the plates were stained with crystal violet solution.

### 4.4. Evaluation of FAK-Copy-Gain and Expression of FAK mRNA and Protein

A modified real competitive PCR (mrcPCR) was performed to detect the *FAK*-copy-gain of breast cancer cell lines as previously described [10].

Reverse-transcription polymerase chain reaction (RT-PCR) and real-time quantitative PCR (real-time qPCR) were performed according to methods described previously [39]. *GAPDH* was used as a control for the level of expression. The primers utilized were as follows: FAK (forward: 5’-TGA GAT CCT GTC TCC AGT CTA CAG-3’, reverse: 5’-CAG TAC CCA TCT ATT AGG TCA GCC-3’), GAPDH (forward: 5’-TGA TGA CAT CAA GAA GGT GGT GAA-3’, reverse: 5’-TCC TTG GAG GCC ATG TGG GCC AT-3’).

Western blotting was performed according to a method described previously [39]. Anti-FAK (#05-537), anti-pFAK (Y397, #05-1140), and anti-cleaved caspase-3 (#AB3623) were purchased from Millipore (Billerica, MA, USA), and anti-β-actin antibody (#A2228) was acquired from Sigma-Aldrich. Anti-Caspase-3 (#9662), anti-AKT (#9272), and anti-pAKT (S473) antibodies were obtained from Cell Signaling (Danvers, MA, USA). 

### 4.5. Acquisition of Databases for the Analysis of the Prevalence of FAK-Copy-Gain in Breast Cancer

The frequency of *FAK*-copy-gain in invasive breast carcinoma patients were analyzed using data from cBioPortal (www.cbioportal.org). The METABRIC (2509 samples) and TCGA (Provisional, 1105 samples) datasets were also analyzed.

### 4.6. Flow Cytometry Analysis for Apoptosis

Flow cytometry was performed according to a method described previously [38]. The Annexin V kit (#51-65874) was purchased from BD Biosciences, and samples were analyzed by the Flow Cytometry Team (National Cancer Center) using FACSVerse (BD Biosciences, San Jose, CA, USA).

### 4.7. Generation of FAK-Conditional Knockdown Cells

Lentiviral doxycycline (Dox)-dependent expression of short harpin RNA (shRNA) was performed according to a method described previously [38]. *FAK*-targeting sequences were inserted into Tet-pLKO-puro vector (Plasmid #21915; Addgene, Cambridge, MA, US). The target sequences were as follows: shFAK #1 (5’-GAT GTT GGT TTA AAG CGA TTT-3’), shFAK #2 (5’-CCG GTC GAA TGA TAA GGT GTA-3’). To knockdown FAK, doxycycline (1 µg/mL) was added. Experiments were carried out in accordance with protocols approved by the National Cancer Center Institutional Biosafety Committee (approval number 17-NCCIBC-039).

### 4.8. RNA Sequencing to Search for Genes Inhibited by FAK Knockdown

Total RNA from doxycycline-treated or untreated cells was extracted with Trizol (#15596018, Life Technologies, Carlsbad, CA, USA) according to the manufacturer’s instructions as reported previously [39]. Preparation of RNA libraries and sequencing were performed by Macrogen (Seoul, Korea) using the HiSeq 2500 sequencing system (Illumina, San Diego, CA, USA). RNA sequencing data were deposited to the Gene Expression Omnibus GEO database under the accession number GSE108596.

Differentially expressed genes (DEGs) were filtered for a fold-change cutoff of 2.0. Genes with less than 1 fragment per kilobase of transcript per million mapped reads (FPKM) in all samples were ruled out. Ingenuity pathway analysis (IPA^®^, QIAGEN, Redwood City, CA, USA) was performed to identify the key biological functions based on curated diseases and functional ontologies in the IPA knowledge database. Heatmapper^®^ software (www.heatmapper.ca) and the Multi-Experiment Viewer 4.9.0 (mev.tm4.org) were used to graphically represent the values.

### 4.9. Tumor Xenograft Experiment on Nude Mice

Female nude mice that were 6 weeks old were purchased from OrientBio (Seongnam, Korea), and maintained in the National Cancer Center Research Institute (NCCRI) animal facility. The animal study protocol had been reviewed and approved by the Institutional Animal Care and Use Committee (IACUC) of NCCRI, which is an Association for Assessment and Accreditation of Laboratory Animal Care International (AAALAC International)-accredited facility, and the animal experiment was performed according to the code of practice for the housing and care of animals bred, supplied or used for scientific purposes (https://www.gov.uk/government/publications/). About 1 × 10^7^ BT549, JIMT, MDA-MB-231, and MDA-MB-453 cells were subcutaneously injected, respectively, into the flank of the mice.

Mice with well-established tumors of about 100 mm^3^ at approximately 10 days postinjection were selected for FAK-inhibitor 14 (F14) treatment. F14 was introduced by intraperitoneal injection at 20 mg/kg, 5 days per week for 4 weeks. Survived mice were sacrificed 5 weeks after the first F14 injection. Tumor size was measured with a caliper and tumor volume in mm^3^ was calculated using the formula (shortest diameter)^2^ × longest diameter/2. Control-group animals were treated with vehicle only. After euthanasia, animals were dissected for removal of tumors and various other organs that were fixed in 4% formaldehyde for routine histology.

### 4.10. Statistical Analysis

Statistical analyses were performed with GraphPad Prism version 5 (GraphPad Software Inc., San Diego, CA, USA). The Mann–Whitney test was used to determine the significance of difference in the IC_50_ values, or the difference in *FAK* RNA expression levels, between copy-gain and non-copy-gain cells. Student’s *t*-test was used to determine the significance of difference in apoptotic percentages or in colony-formation percentages. Two-way repeated-measures ANOVA was used to determine the significance of tumor-volume difference between the untreated and F14-treated groups.

## 5. Conclusions

In conclusion, we demonstrated a significant correlation between *FAK*-copy-gain and FAK-inhibitor sensitivity in breast cancer cells. Especially, considering that several FAK inhibitors are now undergoing clinical trials, our results suggest that *FAK*-copy-gain can be a significant predictive marker for FAK-inhibitor therapy in breast cancer.

## Figures and Tables

**Figure 1 cancers-11-01288-f001:**
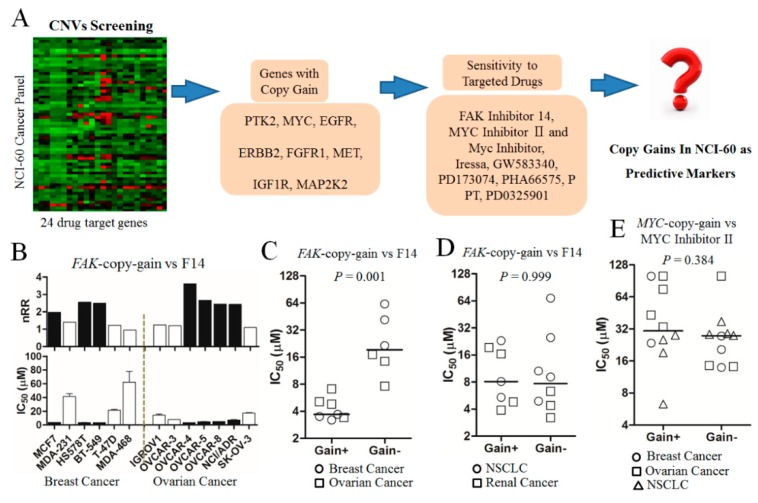
Screening of candidate predictive marker genes. (**A**) Scheme for screening of copy-gain predictive markers. The copy numbers of 24 drug-target genes in an NCI-60 cancer panel had previously been measured by mrcPCR [10]. For eight copy-gain genes in the NCI-60 cells, their targeted drugs were employed for the selection of predictive markers. After determination of IC_50_ for the targeted drugs in cells showing copy gains of target genes, copy-gain predictive markers were analyzed. (**B**) Lower IC_50_ for Focal Adhesion Kinase (FAK) inhibitor 14 (F14) in FAK-copy-gain breast and ovarian NCI-60 cells. Cells with FAK-copy-gain, black bars; cells without copy gain, white bars. nRR: normalized relative ratio of copy number. MDA-231 and MDA-468 in cell line list are MDA-MB-231 and MDA-MB-468, respectively. (**C**) Significantly lower IC_50_ for F14 in cells with FAK-copy-gain in breast and ovarian cancer cells (*p* < 0.001, Mann–Whitney U test). (**D**) No significant correlation between FAK-copy-gain and sensitivity to F14 in non-small-cell lung cancer (NSCLC) and renal cancer cells (*p* = 0.999). For C–D, cells with FAK-copy-gain are marked as Gain+, and those without copy gain as Gain-. (**E**) No significant correlation between MYC-copy-gain and sensitivity to MYC Inhibitor II in breast, ovarian, and NSCLC cancer cells (*p* = 0.384). Cells with MYC-copy-gain are marked as Gain+, and those without copy gain as Gain-.

**Figure 2 cancers-11-01288-f002:**
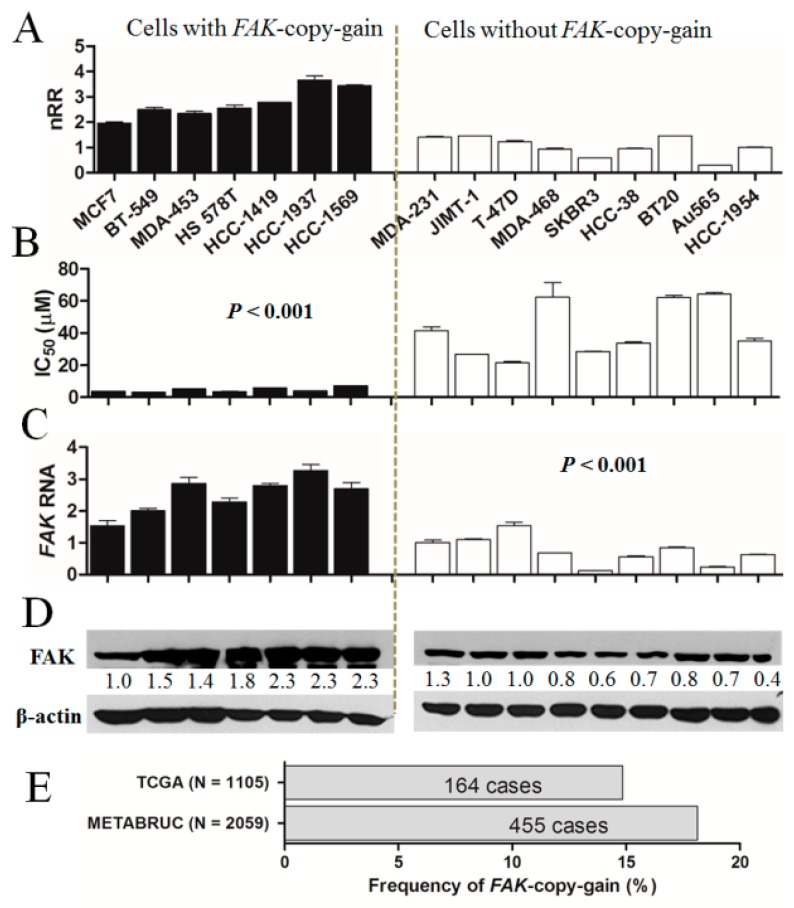
Significant correlation between FAK-copy-gain and sensitivity to F14 in breast cancer cells. (**A**) FAK-copy status of breast cancer cells (*N* = 16). Cells with FAK-copy-gain, black bars; cells without copy gain, white bars. nRR: normalized relative ratio of copy number. MDA-453, MDA-231 and MDA-468 in cell line list are MDA-MB-453, MDA-MB-231 and MDA-MB-468, respectively. (**B**) Significantly lower IC_50_ for F14 in breast cancer cells with FAK-copy-gain (*p* < 0.001, Mann–Whitney U test). Values represent mean ± standard deviation (*N* = 3). (**C**) Significantly higher FAK RNA expression level by real-time qPCR in breast cancer cells with FAK-copy-gain than in those without copy gain (*p* < 0.001). Values represent mean ± standard deviation (*N* = 3). (**D**) Higher FAK expression levels in breast cancer cells with FAK-copy-gain by Western blotting analysis. Representative images of the three independent experiments are shown. The numbers below the blot images indicate the relative expression normalized by β-actin. (**E**) Incidence of FAK-copy-gain in invasive breast carcinoma cases were analyzed from data in cBioPortal (www.cbioportal.org). METABRIC (2509 samples) and TCGA (Provisional, 1105 samples) datasets were analyzed.

**Figure 3 cancers-11-01288-f003:**
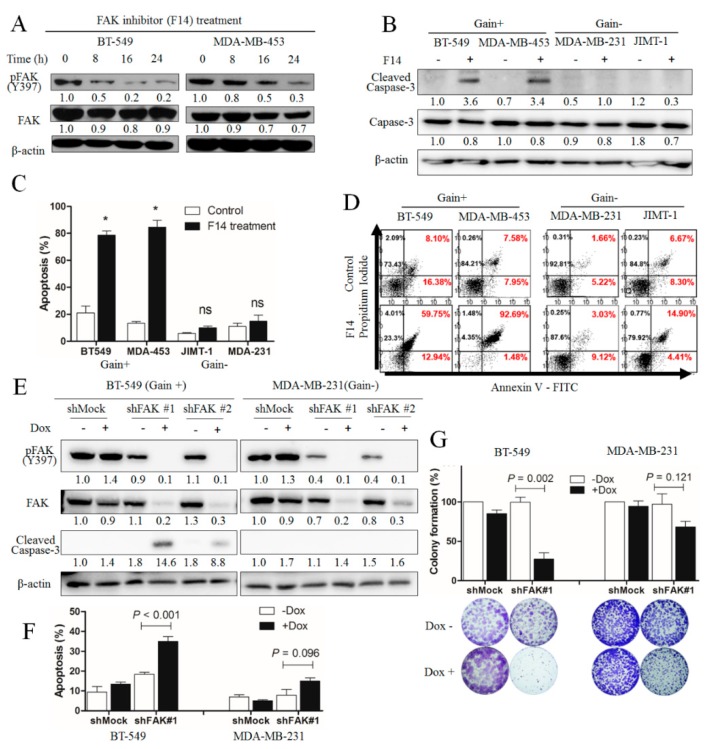
Induction of apoptosis by FAK inhibition in FAK-copy-gain breast cancer cells. (**A**) Decrease of activated or phosphorylated FAK (pFAK) at Y397 by F14 treatment (10 μM) in a time-dependent manner in Western blotting analysis of both BT-549 and MDA-MB-453. (**B**) Specificity of caspase-3 cleavage in FAK-copy-gain cells (Gain+, BT-549 and MDA-MB-453) compared with no FAK-copy-gain cells (Gain-, MDA-MB-231 and JIMT-1) by F14 treatment (10 μM, 6 h) in Western blotting analysis. For A and B, β-actin was used as a loading control. Representative images of the three independent experiments are shown. (**C**) Significantly higher apoptosis by F14 (10 μM, 24 h) in FAK-copy-gain cells (*, P < 0.001) in flow-cytometric analysis. Values represent mean ± standard deviation (*N* = 3). ns, not significant. (**D**) Representative images of flow-cytometric analysis for the percentage of annexin V-positive cells in C. (**E**) Higher caspase-3 cleavage in BT-549 by induction of FAK shRNA (shFAK #1 and shFAK#2) on Western blot analysis. FAK, FAK phosphorylation (pFAK), and caspase-3 cleavage (Cleaved Capase-3) were measured by Western blot. β-actin was used as a loading control. Representative images of the three independent experiments are shown. MDA-453 and MDA-231 in the cell line list are MDA-MB-453 and MDA-MB-231, respectively. (**F**) Flow-cytometric analysis of apoptosis by FAK knockdown. A significantly higher apoptotic fraction was observed—using flow-cytometric analysis—in BT-549 (*p* < 0.001) than in MDA-MB-231 (*p* = 0.096) by induction of FAK shRNA (shFAK#1). The percentage of annexin V-positive cells is indicated as the mean ± standard deviation (*N* = 3). (**G**) A significantly higher reduction in colony formation (*p* = 0.002) was observed in BT-549 than in MDA-MB-231 (*p* = 0.121) as a result of FAK shRNA induction (shFAK#1). The lower figures represent the representative colonies with or without induction with doxycycline in BT-549 or MDA-MB-231 cells transfected by either mock shRNA (shMock) or FAK shFAK. Values represent mean ± standard deviation (*N* = 3). Representative images and the relative percentages of colonies are shown. For E–G, breast cancer cells stably expressing doxycycline-inducible FAK shRNAs (shFAK#1 or shFAK#2) or shMock were treated with (+Dox, 1 μg/mL) or without doxycycline (-Dox). After induction of shRNA for 6 days (E and F) or 15 days (G), Western blot, flow cytometry, and crystal violet staining were performed. For G, cells were seeded at a low density. ns, not significant (*p* > 0.05). The numbers below the blot images in A, B, and E indicate the relative expression normalized by β-actin.

**Figure 4 cancers-11-01288-f004:**
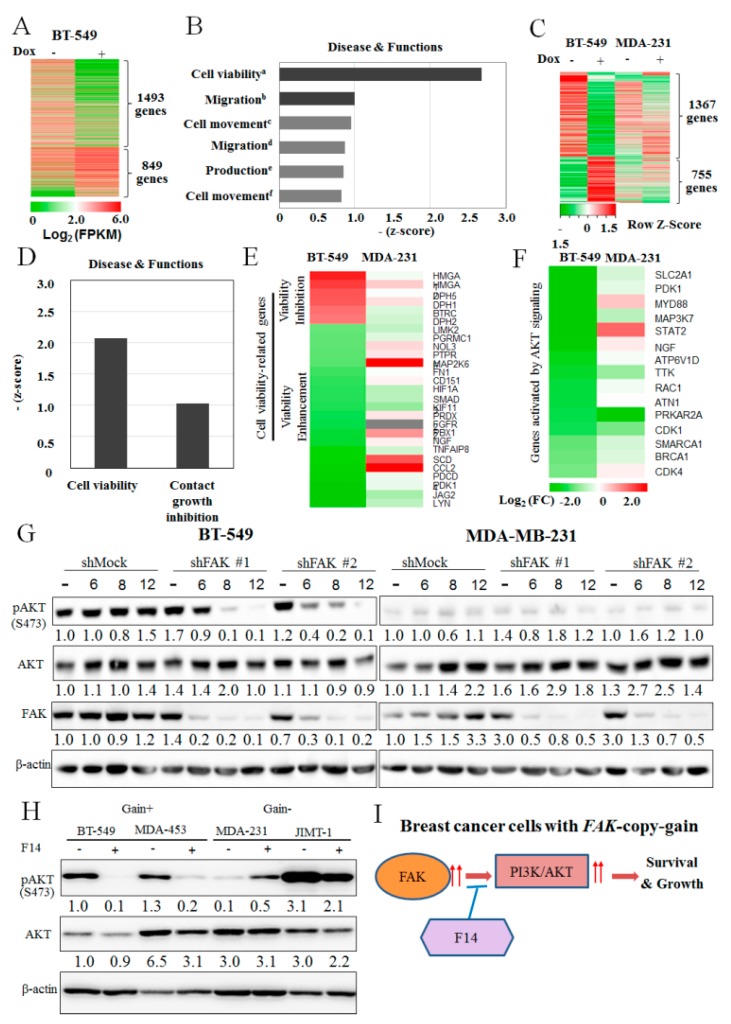
Downstream molecular pathways in FAK-copy-gain cancer cells by FAK knockdown. (**A**) Differentially expressed genes (DEGs) by RNA sequencing in BT-549 Tet-on-shFAK cells with (Dox, +) or without doxycycline treatment (Dox, -) for 6 days. Genes whose expression changed more than two-fold are shown. The gene list is presented in Appendix A. (**B**) Major pathways in DEGs by core analysis using IPA. Suppressed signal pathways by FAK knockdown were related to cell viability and migration. The pathways with more than 0.8 absolute values of z-score are shown. a; cell viability of breast cancer cell lines, b; migration of tumor cell lines, c; cell movement of tumor cell lines, d; migration of breast cancer cell lines, e; production of reactive oxygen species (ROS), f; cell movement of breast cancer cells. (**C**) Differential DEG patterns between BT-549 and MDA-MB-231 by FAK knockdown. The gene list is also presented in Appendix A. (**D**) Analysis of BT-549-specific DEGs by IPA. The most suppressed BT-549-specific pathways were related to cell viability rather than to cell migration or movement by FAK knockdown. (**E**) Gene list of BT-549-specific DEGs. Among the BT-549-specific DEGs, genes related to inhibition of viability were upregulated and those associated with increased viability were downregulated by knockdown of FAK. (**F**) BT-549-specific downregulation of genes was activated by AKT signaling by knockdown of FAK. AKT downstream genes were consistently downregulated in BT-549 cells by FAK knockdown but not in MDA-MB-231 cells. The color scales of DEGs for E and F are the same. (**G**) BT-549-specific reduction of AKT phosphorylation by FAK knockdown on Western blot analysis. FAK knockdown reduced AKT phosphorylation in BT-549 cells but not in MDA-MB-231 cells. Cells stably expressing doxycycline-inducible FAK shRNAs (shFAK#1 or shFAK#2) or mock shRNA (shMock) were treated with (+Dox, 1 μg/mL) or without doxycycline (-Dox) for 6 days. AKT, AKT phosphorylation (pAKT), and FAK were analyzed by Western blotting analysis. β-actin was used as a loading control. (**H**) Reduction of AKT phosphorylation by F14 treatment in FAK-copy-gain cells. After F14 treatment (10 μM, 6 h), AKT, AKT phosphorylation (pAKT), and FAK were analyzed. AKT phosphorylation was reduced only in FAK-copy-gain cells (BT-549 and MDA-MB-453) but not in cells without FAK-copy-gain (MDA-MB-231 and JIMT-1). MDA-453, MDA-231 and MDA-468 in cell line list are MDA-MB-453, MDA-MB-231 and MDA-MB-468, respectively. (**I**) Proposed model. In breast cancer, cells with FAK-copy-gain are dependent on FAK activity for activation of PI3K/AKT signaling and cell survival. The effect of the FAK inhibitors can be more effective when treating breast cancer patients with FAK-copy-gain. The numbers below the blot images in G and H indicate the relative expression normalized by β-actin.

**Figure 5 cancers-11-01288-f005:**
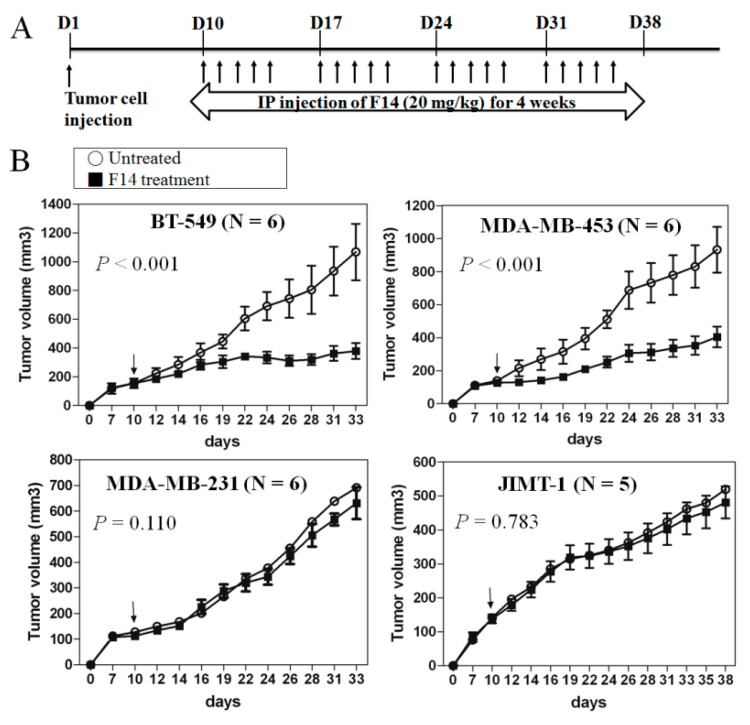
Effect of F14 in FAK-copy-gain breast cancer cells in a mouse xenograft model. (**A**) Scheme of experimental method. Once tumors were well-established after subcutaneous cancer cell injection (Day 10), F14 (20 mg/kg) was injected 5 days per week for 4 weeks. (**B**) Specific inhibition of tumor growth in FAK-copy-gain cells by F14 treatment in a mouse xenograft model (*p* < 0.001 for both BT-549 and MDA-MB-453, by two-way RM ANOVA). Tumor volumes on the indicated days were determined as described in the Materials and Methods section. Values represent mean ± standard deviation (*N* = 5 for JIMT-1 and *N* = 6 for the other cells). ns, *p* > 0.05.

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
