# Peer review of "FAK-Copy-Gain Is a Predictive Marker for Sensitivity to FAK Inhibition in Breast Cancer"

_cancers, 2019, doi:10.3390/cancers11091288_

Round 1

Reviewer 1 Report

The authors present an interesting finding whereby FAK copy number gain could be developed into a biomarker for breast cancer. Clearly there is some limitation o this finding, because the authors indicate this is useful for breast cancers that harbor robust FAK-signalling.

Since not all breast cancers express FAK, and given ER-/TNBC tumors (ones that typically contain higher activity of MAPK / FAK signalling) are a rarer form of tumor, the usefulness of this test as a biomarker may be limited. 

Have the authors assessed any other scenario in which FAK copy gain associates with a positive killing effect of FAK inhibition in culture. or for that matter is FAK inhibition not effective as a therapeutic in ER+ cells (MCF-7), or steroid hormone refractory lines 

please address these concerns and resubmit.

Author Response

Reply: FAK expression is reported to be higher in TN breast cancer, which is about 10-20% of total breast cancer cases (Golubovskaya et al., 2014). However, FAK expression is not limited to TN breast cancer, as reported in that paper. Therefore, target patients need not be limited to TN breast cancer cases. Our analysis on the TCGA and METABRIC datasets also indicates that about 15% of breast cancer patients can be candidates for FAK inhibitor treatment, as is consistent with another previous study reporting that about 18% of breast cancer tissues showed high polysomy or amplification of FAK (Yom et al., 2011).

The fact of higher FAK expression in TN breast cancer has been added to the Discussion of our manuscript (line 324).

Reviewer 2 Report

I’ll begin by saying that I am not familiar with FAK, FAK inhibitors, or their relevance to (breast) cancer, and so cannot comment on how the current study sits within the body of literature in this field. But I do evaluate small molecules as cancer therapeutics, and it is upon this basis that I review the current manuscript.

In this study, Kim et al seek to identify new predictive markers for targeted therapies with the central premise that copy-number gains in anti-cancer targets will yield increased sensitivity to the corresponding inhibitor of that target. Interrogating copy-number gains and drug sensitivities in the NCI-60 cell line panel, they identify Focal Adhesion Kinase (FAK) and FAK inhibitor F14 as one such example within a subset of cell lines (breast and ovarian). Focusing upon breast cancer cell lines, they show that those with FAK copy-gain have increased mRNA and protein expression, whilst those without FAK copy-gain do not, and accordingly, show that those with FAK copy-gain are more sensitive to FAK inhibitor F14. Continuing with mechanistic experiments, the authors show that F14 treatment induces apoptosis in FAK-copy gain breast cancer cell lines but not in breast cancer cells without FAK copy-gain, and in line with this, using shRNA knockdown of FAK, they show FAK-copy gain cells also undergo apoptosis and inhibition of proliferation, whilst this is not observed in cells without FAK copy-gain. Analysis of gene expression in FAK knockdown cells implicates Akt signalling in the differential response observed. And finally, the authors show heterotopic xenograft mouse experiments highlighting the anti-cancer activity of F14 against FAK copy-gain tumours over tumours without FAK copy-gain.

Altogether, this is a nice study with a consistent story and the data presented supports the conclusions made by the authors – FAK copy-gain could be a predictive marker for sensitivity to FAK inhibitor F14, at least within cultured cells.

I have some comments.

There is no introduction to FAK and the inhibitor used throughout the study – this should be included in the introduction.

Sections 2.1. and 2.2. refer to “correlations” however no correlation analyses have not been performed. The authors have separated cell lines into groups (with gain, without gain) and showed that there is differential FAK expression, F14 IC50 etc between these groups, but there is no correlation analyses of these data points. Would suggest either performing such correlations (XY plot of copy-gain/mRNA/protein versus IC50 with subsequent pearson/spearman correlation analysis) or re-word these sections to reflect what is shown.

In Figure 2, FAK mRNA and protein expression is compared in a cell line panel, however, adequate controls are lacking here. For the Q-PCR analysis, one reference gene is chosen – GAPDH – whilst current guidelines recommend at least two validated reference genes be used. The problem is, of-course, that expression of the reference gene can fluctuate between cell lines, making results inaccurate – which is discussed in the linked paper - https://www.ncbi.nlm.nih.gov/pubmed/26555275. Along these lines, a similar problem exists with Western blot analysis of a cell line panel, and the ideal control would be ponceau staining of the membranes to confirm equal loading.

Figure 5 could be complemented with further data form the performed experiment which could be of interest to readers, such as body weight measurements from the treated mice.

This is just a suggestion of how the current dataset could be complemented and expanded (i.e. this is not a requirement) – you could interrogate the Achilles dataset from Broad (depmap.org), breast cancer cell lines could be split between those with and without FAK-copy gain, and the dependency of FAK upon survival of these cell lines (from both the RNAi dataset and the CRISPR dataset) could be analysed. Could be a nice addition.

Some other comments:

Lines 261-263 – these should not be here…

Line 270 – concluding sentence – “…FAK copy-gain is asignificant predictive marker … in breast cancer.” Would suggest replacing “is a” to “could be” as the current study is performed in cultured cells and here you are concluding about breast cancer.

Lines 272 – 276 – I got a little confused here, suggest revising sentence structure. This also highlights one point worth adding to your manuscript, you are assume throughout that targeted inhibitors work through their intended targets, however there are many examples in the literature of compounds being off-target. And this could explain, in some instances at least, why copy-number gains of gene X to not yield sensitivity to inhibitor of X. It’s a small point perhaps worth including.

Round 2

Reviewer 1 Report

please make minor grammatical errors to the text

Author Response

August 20, 2019

Dr. Molly Li

Assisant Editor, MDPI

Dear Dr. Li,

Thank you for the reviewers’ indispensable comments. The manuscript has been improved as a result. Please find the revised manuscript.

Grammar errors in our manuscript have been edited. Fig. 2 has been changed, due to modification of the term ‘FAK-copy gain’ into ‘FAK-copy-gain’.

Thank you.

Sincerely,

Kyeong-Man Hong, MD, PhD

Research Institute, National Cancer Center

Goyang 10408, Republic of Korea

Tel: +82-31-920-2261

Hyoncheol Jang, PhD

Research Institute, National Cancer Center

Goyang 10408, Republic of Korea

Tel: +82-31-920-2239
